# Stress granules are not present in Kras mutant cancers and do not control tumor growth

Maxime Libert[1], Sophie Quiquempoix[1], Jean S Fain[1], Sébastien Pyr dit Ruys [1], Malak Haidar[1], Margaux Wulleman[1], Gaëtan Herinckx [1], Didier Vertommen [1], Christelle Bouchart [2], Tatjana Arsenijevic[3], Jean-Luc Van Laethem[3] & Patrick Jacquemin [1]✉

## Abstract

**Stress granules (SG) are membraneless ribonucleoprotein-based cytoplasmic organelles that assemble in response to stress. Their formation is often associated with an almost global suppression of translation, and the aberrant assembly or disassembly of these granules has pathological implications in neurodegeneration and cancer. In cancer, and particularly in the presence of oncogenic KRAS mutations, in vivo studies concluded that SG increase the resistance of cancer cells to stress. Hence, SG have recently been considered a promising target for therapy. Here, starting from our observations that genes coding for SG proteins are stimulated during development of pancreatic ductal adenocarcinoma, we analyze the formation of SG during tumorigenesis. We resort to in vitro, in vivo and in silico approaches, using mouse models, human samples and human data. Our analyses do not support that SG are formed during tumorigenesis of KRAS-driven cancers, at least that their presence is not universal, leading us to propose that caution is required before considering SG as therapeutic targets.**

**Keywords** Cancer; KRAS; PDAC; Stress Granules
**Subject Categories** Cancer; RNA Biology; Signal Transduction

## Introduction

Stress granules (SG) are dense membraneless aggregates composed of proteins and RNA found in the cytoplasm of stressed cells. Various stresses, such as heat shock, nutrient deprivation, or the presence of reactive oxygen species, can induce their formation (Glauninger et al, 2022; Ripin and Parker, 2023). The composition of cytoplasmic SG, whether their protein or RNA content, varies according to the conditions of their formation. A compilation of different studies indicates that hundreds of proteins, a significant part of which are defined as RNA binding proteins (RBP), can be integrated into SG (Youn et al, 2019). Among these proteins, some are essential for the formation of SG, as in their absence, SG cannot

be formed; this is particularly the case for the stress granule assembly factors Ras GTPase-activating protein SH3 domain-binding protein (G3BP)1, G3BP2, and ubiquitin-associated protein 2 like (UBAP2L) (Matsuki et al, 2013; Markmiller et al, 2018).

The function of SG is still subject to discussion, and many hypotheses have been put forward to explain their role. A recurring hypothesis is that they allow transiently protecting transcripts from degradation during stress. This hypothesis stems in part from the observation that the presence of stress granules is associated with an overall inhibition of protein synthesis, with the notable exception of some transcripts encoding proteins directly involved in the management of a stressful stimulus, such as heat shock proteins, which are not integrated into SG and are highly translated during stress (Glauninger et al, 2022). These results were largely obtained from cultured cells treated with arsenite, a powerful inducer of SG, generating oxidative stress. In vivo, whether in animal models or in humans, pathologies where the presence of SG has been described are less common. Currently, these pathologies are mainly linked to the nervous system or to cancers driven by the KRAS oncogene. In the nervous system, mutations in DEAD-box RNA helicases that induce SG formation and concomitant inhibition of translation are found in neurodevelopmental diseases and brain cancer (Valentin-Vega et al, 2016; Lessel et al, 2017; Lennox et al, 2020). For KRAS-driven cancers, two seminal studies have greatly contributed to popularizing this field (Redding and Grabocka, 2023). A study showed that development of SG is markedly elevated in mutant KRAS cells and tumors. Mechanistically, mutant KRAS upregulates SG formation by stimulating the production of prostaglandin 15-d-PGJ2 which enables paracrine control of SG, eventually resulting in resistance of cancer cells to stress stimuli (Grabocka and Bar-Sagi, 2016). Another study stated that obesity, as a stressful condition, is a driver of SG formation in pancreatic ductal adenocarcinoma (PDAC) and that PDAC growth depends on SG (Fonteneau et al, 2022). These two studies concluded that SG are key determinants of tumorigenesis and that inhibiting SG formation will block tumor growth.

Our interest in SG arose from the analysis of transcriptomic data showing that the expression of several mRNAs encoding SG proteins is increased during pancreatic cancer initiation. However, our further analyses do not support that SG are formed in KRAS-driven cancers.

[1]Université catholique de Louvain, de Duve Institute, 1200 Brussels, Belgium. [2]Department of Radiation Oncology, Jules Bordet Institute, Brussels, Belgium. [3]Université libre de Bruxelles, Erasme University Hospital, Laboratory of Experimental Gastroenterology, Brussels, Belgium. ✉E-mail: patrick.jacquemin@uclouvain.be

# Results and discussion

## Pancreatic cancer initiation is associated with an increased expression of mRNA encoding SG components

Cerulein-induced inflammation of pancreas expressing an oncogenic form of KRAS in acinar cells leads to the development of preneoplastic lesions called Pancreatic Intraepithelial Neoplasia (PanIN) (Guerra et al, 2011). In our previous study (Assi et al, 2021), we characterized the transcriptome of purified pancreatic acinar cells expressing oncogenic KRAS$^{G12D}$, in the absence or presence of inflammation, after repeated injections of cerulein in tamoxifen-treated adult ElastaseCre$^{ER}$ LSLKras$^{G12D}$ (ElaK) mice. Here, we found that out of the 1747 significantly upregulated mRNA (p-value <0.05; log$_2$ fold change >0.5) in the presence of inflammation, 97 encode stress granule (SG) proteins. This represents almost 20% of the 488 proteins identified as components of these granules with SG score ≥4 (Youn et al, 2019). mRNA encoding RBP, listed in the RBD database (http://rbpdb.ccbr.utoronto.ca/), represented a significant part (26%) of the 97 upregulated mRNA (Fig. EV1 and Dataset EV1). At this stage, we considered that this increase could be associated with the presence of SG in PanIN (Grabocka and Bar-Sagi, 2016).

## SG are not observed in mouse models of Kras mutant tumors

We next sought to detect the presence of proteins essential for the formation of SG, namely G3BP1, G3BP2, and UBAP2L, by immunolabeling of PanIN in cerulein-treated ElaK mice. Prior to conducting these experiments, we validated the specificity of the selected antibodies. Antibodies recognizing human G3BP1, G3BP2, and UBAPL2 were tested on PFA-fixed and paraffin-embedded HeLa cells, after treatment or not with arsenite (Fig. EV2A). The results showed the expected labeling of cytoplasmic proteins in untreated and arsenite-treated cells. To confirm the specificity of antibodies against mouse G3BP1, a Myc-tagged mouse G3BP1 expression vector was transfected into HeLa cells. Western blotting indicated that 2 of the 3 tested antibodies recognized mouse G3BP1 (Fig. EV2B).

We then used these antibodies on mouse pancreas sections containing PanIN. We observed increased expression of G3BP1 in PanIN, as compared to metaplastic acini. However, G3BP1 was uniformly distributed in the cytoplasm of PanIN and did not show granular expression, suggesting the absence of SG (Fig. 1A). A similar result was obtained for G3BP2 (Fig. 1B), while UBAP2L was not detected in PanIN (Fig. 1C). Likewise, G3BP1 and G3BP2 were detected, but no SG was observed in PanIN from ElaCre$^{ER}$ LSL-KRas$^{G12D/+}$ LSL-Trp53$^{R172H/+}$ (ElaKP) mice or in tumors arising from orthotopic graft of PDAC cells generated from ElaKP mice (Fig. 1D,E). To ensure that our experimental setup could detect the presence of SG, we subjected a mouse pancreas with PanIN to a heat shock. The pancreas was taken out of the abdominal cavity and incubated in phosphate-buffered saline (PBS) for 30 min at 37 °C or 43 °C, and subsequently immunolabeled. This revealed the presence of SG in the PanIN after heat shock (Fig. 1F). We concluded that SG are not detectable in mouse PanIN, under the usual induction conditions, namely by injection of cerulein, in the absence of heat shock. Our data are in contrast with published data which

illustrated the presence of SG in PanIN (Grabocka and Bar-Sagi, 2016; Fontenau et al, 2022). We noted that in Grabocka and Bar-Sagi (2016), the description of SG in mouse PanIN relied on a monoclonal anti-human G3BP1 antibody which we found incapable of recognizing mouse G3BP1, unlike the two other G3BP1 antibodies tested (Fig. EV2B).

We also tested the presence of SG in another model of tumorigenesis induced by KRAS$^{G12D}$, namely the induction of lung tumors using intratracheal instillation of adenoviruses expressing Cre recombinase in LSL-KRas$^{G12D/+}$ mice (DuPage et al, 2009; Jackson et al, 2001). Three months after instillation, we analyzed the pulmonary adenomas but did not detect SG using antibodies against G3BP1, G3BP2, and UBAP2L, as well as CAPRIN1, another SG protein (Fig. EV3).

## The protein interactome of G3BP in PanIN does not support a role for G3BP linked to SG

To explore the extent to which the protein interactome of G3BP in PanIN differs from that found in SG, we independently immunoprecipitated G3BP1 and G3BP2 from pancreas extracts of control or cerulein-treated ElaK mice and determined the interactomes by mass spectrometry. In the pancreas from control and cerulein-treated mice, as expected, G3BP1 was found in the corresponding immunoprecipitation (Fig. 2A). In addition, G3BP1 was found in the G3BP2 interactome and conversely, G3BP2 was found in the G3BP1 interactome (Dataset EV2). The interactome of G3BP1 in Hela cells treated with arsenite was also obtained; a comparison of this interactome with published data allowed us to validate our immunoprecipitation protocol: among 106 co-immunoprecipitated proteins, 42 were SG proteins (Fig. 2B, Dataset EV2). Interestingly, in the control ElaK pancreas, 28 proteins found in SG (Youn et al, 2019) were present in the G3BP1 and G3BP2 interactome, while in the cerulein-treated pancreas with PanIN, only 22 SG proteins were present in the G3BP1 and G3BP2 interactome (Fig. 2C, Dataset EV2). As anticipated, UBAP2L was not present in this interactome, consistent with its absence in PanIN. The small proportion of SG proteins found in the G3BP1/2 interactome in the presence of cerulein suggests SG are not present in PanIN. Together, these results do not support that G3BP proteins in PanIN are included in SG.

## Mutant KRAS does not exert cell non-autonomous control on SG

Previous data (Grabocka and Bar-Sagi, 2016) claim that SG confer cytoprotection against stress stimuli and chemotherapeutic agents in mutant KRAS cancers. This study argues that SG are markedly elevated in arsenite-treated mutant KRAS cell lines compared to arsenite-treated wild-type KRAS cell lines; a SG index, which measures the surface occupied by SG in cells, was thus 12 times higher in the former. To verify this result, we decided to compare several cancer cell lines harboring or not a mutant version of KRAS, and first performed a search on PubMed with "stress granules cell lines" (sorted by Best Matches on 03/14/2023). We reasoned that with such a difference, mutant KRAS cell lines should have been historically selected by SG researchers for their studies. We found that 17 out of 21 cell lines used in the top 10 articles are wild-type KRAS. Since this does not suggest that mutant KRAS cells have

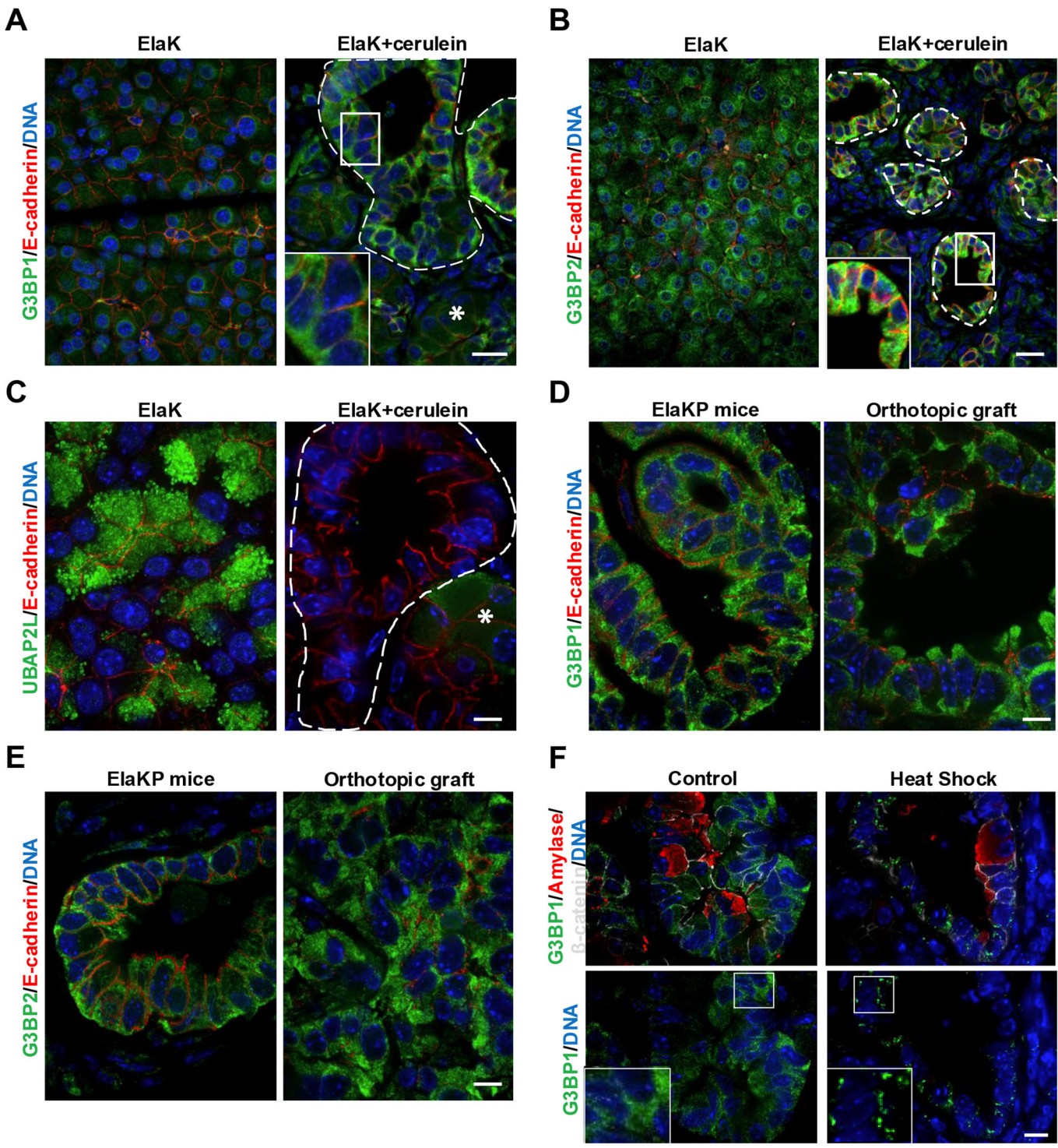

**Figure 1. SG are not present in PanIN and PDAC.**

(A) Sections from pancreas of ElaK mice treated with cerulein or not (control) immunolabelled with G3BP1 and E-cadherin (plasma membrane marker). (B) Sections from pancreas of ElaK mice treated with cerulein or not (control) immunolabelled with G3BP2 and E-cadherin (plasma membrane marker). (C) Sections from pancreas of ElaK mice treated with cerulein or not (control) immunolabelled with UBAP2L and E-cadherin. (D) Sections from pancreas of ElaK mice and from orthotopic tumors generated with ElaKP-derived PDAC cells immunolabelled with G3BP1 and E-cadherin. (E) Sections from pancreas of ElaKP mice and from orthotopic tumors generated with ElaKP-derived PDAC cells immunolabelled with G3BP2 and E-cadherin. (F) Sections from cerulein-treated ElaK pancreas shocked at 37 °C (control) or 43 °C (heat shock) during 30 min in PBS immunolabelled with G3BP1, amylase (acinar cell marker) and β-catenin (plasma membrane marker). At least three mice per group were used. Nuclei were counterstained with Hoechst. *: metaplastic acinar cell. Dotted lines: PanIN. Scale bar: 20 µm (A, B) and 10 µm (C–F). The pictures shown are representative of tissues from at least 3 biological replicates. Source data are available online for this figure.

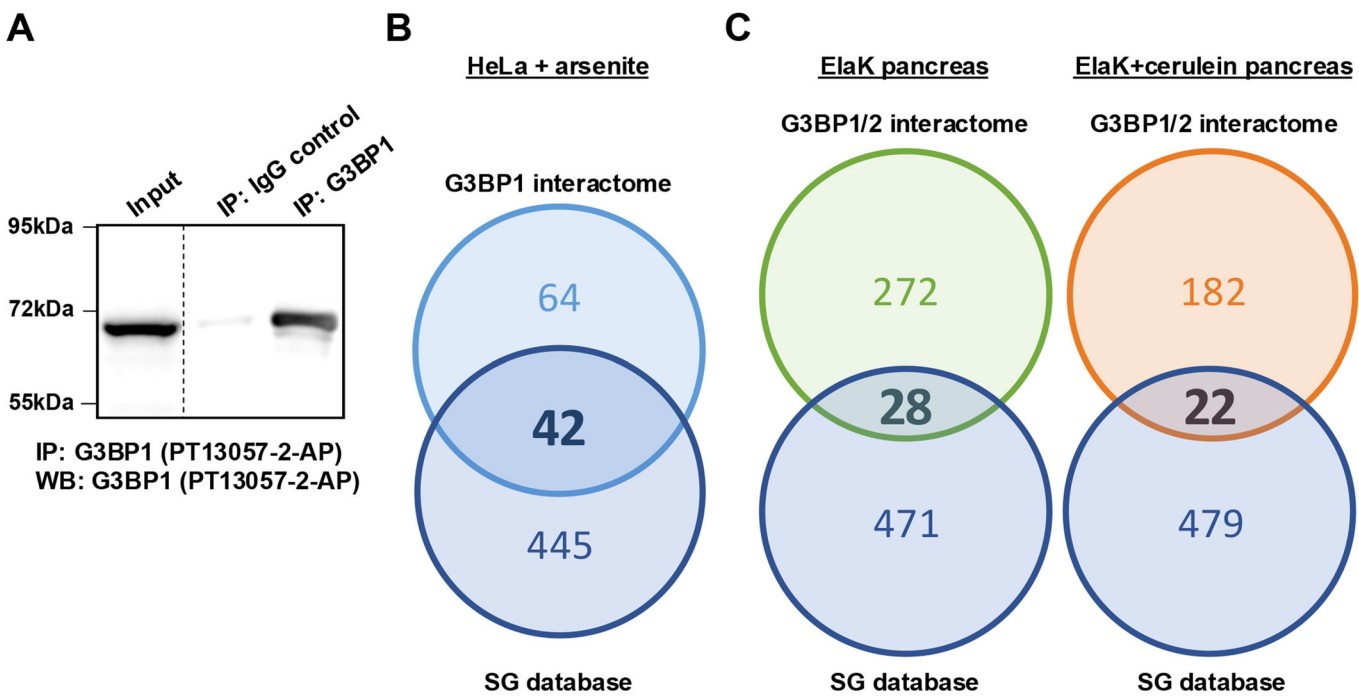

**Figure 2.   Few SG proteins are present in the proteins interacting with G3BP in PanIN.**

(A) Western blotting performed with G3BP1 antibody (13057-2-AP) on protein extracts from Hela cells treated with arsenite. Immunoprecipitation (IP) was performed with the same antibody. The pictures shown are representative of tissues from at least 3 biological replicates. (B) Venn diagram showing the number of common proteins among proteins found in the SG database and proteins associated with G3BP1 in HeLa cells treated with arsenite. (C) Venn diagram showing the number of common proteins among proteins found in the SG database and proteins associated with G3BP1/2 in an ElaK (left) and cerulein-treated ElaK (right) mouse pancreas. Source data are available online for this figure.

larger SG than cells with wild-type KRAS, we subjected wild-type and mutant KRAS cell lines to arsenite treatment. SG were detected by G3BP1 and CAPRIN1 immunolabelling. The calculated SG index revealed no correlation between the SG index and the presence of KRAS mutations (Fig. 3A).

It was also stated that mutant KRAS upregulates SG by stimulating the production of prostaglandin 15-d-PGJ2, and that 15-d-PGJ2 obviates the requirement of mutant KRAS for upregulation of SG in wild-type KRAS lines, indicating that mutant KRAS exerts cell non-autonomous control on SG index through 15-d-PGJ2 (Grabocka and Bar-Sagi, 2016). To verify this conclusion, we transfected vectors expressing for GFP, wild-type KRAS, or mutant KRAS in wild-type KRAS HEK293 and HeLa cells, and measured the SG index. While a 5-fold increase in SG index was previously observed in HeLa cells in the presence of mutant KRAS (Grabocka and Bar-Sagi, 2016), we observed a 1.5-fold increase in SG index in HEK293 cells and no significant increase in HeLa cells (Fig. 3B). Also, we did not detect any 15-d-PGJ2 increase in the culture medium of HEK293 cells transfected with mutant KRAS, as compared to that of cells transfected with GFP or wild-type KRAS (Fig. 3C). Finally, we treated wild-type KRAS cells with 15-d-PGJ2, as previously described (Grabocka and Bar-Sagi, 2016). We observed no difference in the SG index between untreated and prostaglandin-treated cells, with the exception of a slight increase observed in BXPC3 cells (Fig. 3D).

A connection between mutant KRAS and 15-d-PGJ2 has also been established in human cancer patients. Indeed, mutant KRAS

was involved in the metabolism of 15-d-PGJ2 by increasing *PTGS2* mRNA (COX-2, the enzyme producing 15-d-PGJ2) and by decreasing *HPGD* mRNA (the enzyme catabolizing 15-d-PGJ2); it was notably observed a significant enrichment of *PTGS2* in lung adenocarcinoma (LUAD) harboring KRAS mutations, compared to LUAD bearing wild-type KRAS (Grabocka and Bar-Sagi, 2016). We repeated the same analysis by including a larger number of patients (502 vs 24) and were unable to confirm that KRAS mutations has opposite effects on the expression of these two enzymes in LUAD. Similar conclusions were drawn from our analysis of colon adenocarcinoma (COAD) and PDAC, two other cancers with highly prevalent KRAS mutations (Fig. 3E). Thus, our results do not support that mutant KRAS is associated with changes in enzyme expression that would lead to increased 15-d-PGJ2 production, and that 15-d-PGJ2 upregulates SG.

We further investigated whether the presence of KRAS mutations was associated with increased expression of mRNA encoding SG proteins. This was not the case in PDAC, LUAD, and COAD, as we observed no correlation between the number of upregulated SG mRNA when comparing the total number of upregulated mRNA in cancers with KRAS mutations versus cancers without KRAS mutations (Fig. 4A–D). In LUAD, the number of downregulated mRNA encoding SG proteins was even much higher than the number of upregulated mRNA encoding SG proteins. All together, these results do not establish any link between a KRAS mutation and the presence of SG.

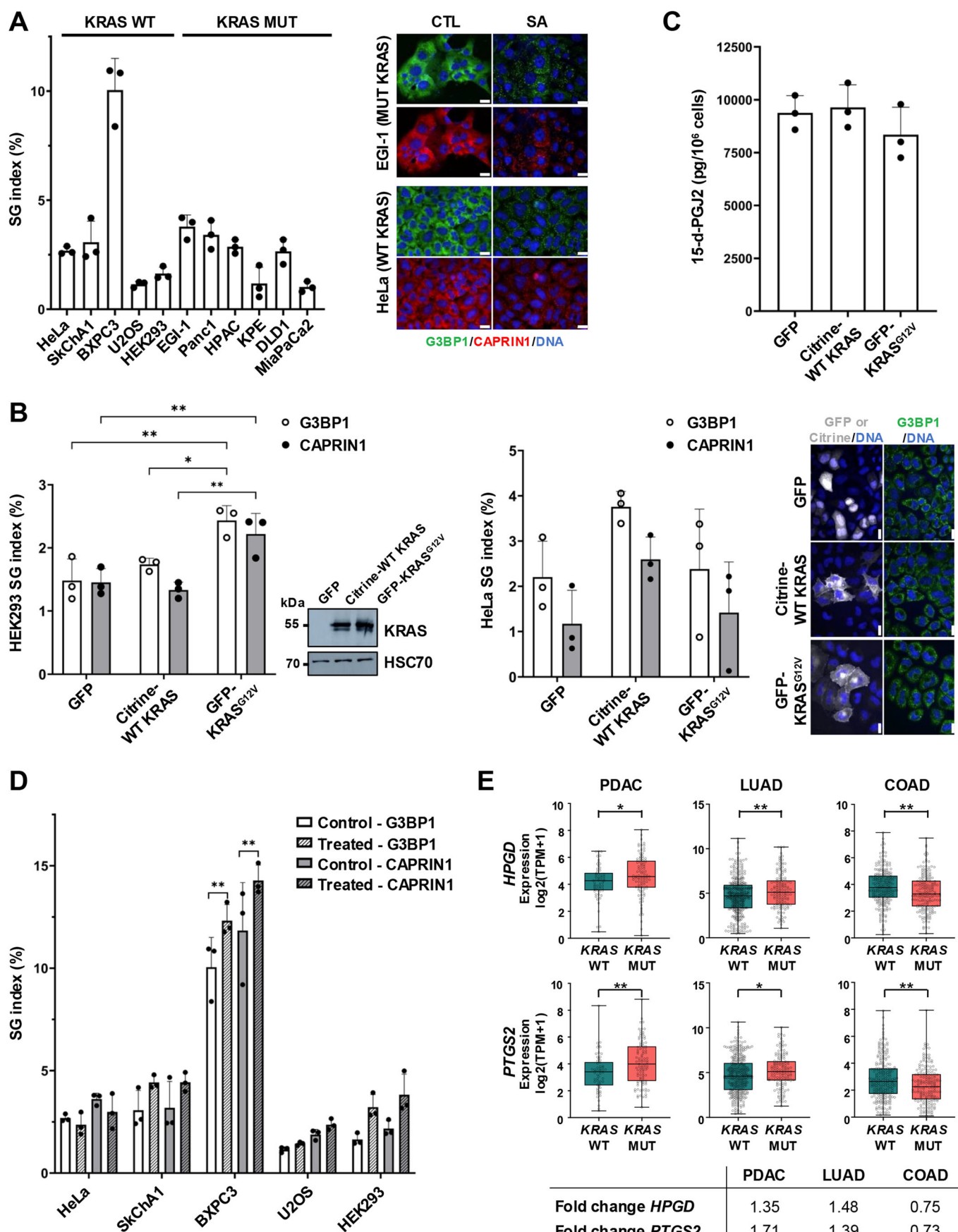

**Figure 3.   Kras mutation is not associated with SG formation.**

(A) SG index of five wild-type and six mutant KRAS cell lines subjected to arsenite treatment to induce cytoplasmic SG formation. SG were detected by G3BP1 and CAPRIN1 immunolabelling and quantified by calculating the SG index (SG surface divided by cell surface). The data are presented as the percentage of SG surface relative to the total cell surface. Representative pictures of G3BP1 and CAPRIN1 immunolabeling obtained on EGI-1 cells (mutant KRAS) and HeLa cells (wild-type KRAS) are shown. Scale bar: 20 μm. Only the SG index quantified from G3BP1 labeling is shown due to space constraints. (B) SG index of HEK293 (left) and HeLa (right) cells transfected with plasmids encoding GFP, Citrine-wild-type (WT) KRAS and GFP-KRAS$^{G12V}$, and treated with arsenite. SG index was calculated as described in (A). No statistically significant differences were observed for the SG index in HeLa cells. To confirm transfection efficiency, whole lysates of HEK-293 cells were collected and subjected to Western blotting with anti-KRAS antibody, HSC70 serving as a loading control. Representative pictures of GFP, Citrine, and G3BP1 immunolabeling obtained on HeLa cells are shown. Scale bar: 20 μm. (C) Levels of 15-d-PGJ2 secretion measured in the culture medium of HEK293 cells transfected as described in (B). No statistically significant differences were observed in 15-d-PGJ2 levels. (D) SG index of wild-type KRAS cell lines subjected to arsenite treatment to induce cytoplasmic SG formation, and cultured in the absence (Control) or presence (Treated) of 15-d-PGJ2. SG were detected by G3BP1 and CAPRIN1 immunolabelling and quantified by calculating the SG index (SG surface divided by cell surface). The data are presented as the percentage of SG surface relative to the total cell surface. Only statistically significant differences are indicated. (E) mRNA expression levels of *HPGD* (top) and *PTGS2* (bottom) in three human cancers (colon adenocarcinoma-COAD, lung adenocarcinoma-LUAD, and pancreatic ductal adenocarcinoma-PDAC) categorized by Kras mutation status. Each dot represents one patient. The y-axis represents RNA-Seq by expression log$_2$(TPM + 1). The lower panel shows the fold changes between mean expressions in *KRAS* WT and *KRAS* mutant patients. The box plots display the minimum, 25th percentile, median, 75th percentile, and maximum. (A–D) Data presented are from a representative experiment out of at least 3 technical replicates leading to the same conclusion. Quantifications were performed on 10 randomly acquired images at 40× magnification for each cell line. Data are presented as means ± standard error of the mean (SEM). (B–E) For single comparisons between two experimental groups, an unpaired Student's t-test was performed. To identify significant differences between multiple groups, data were subjected to a one-way or two-way ANOVA with Tukey's correction for multiple comparisons. *$p < 0.05$; **$p < 0.01$; ***$p < 0.001$.

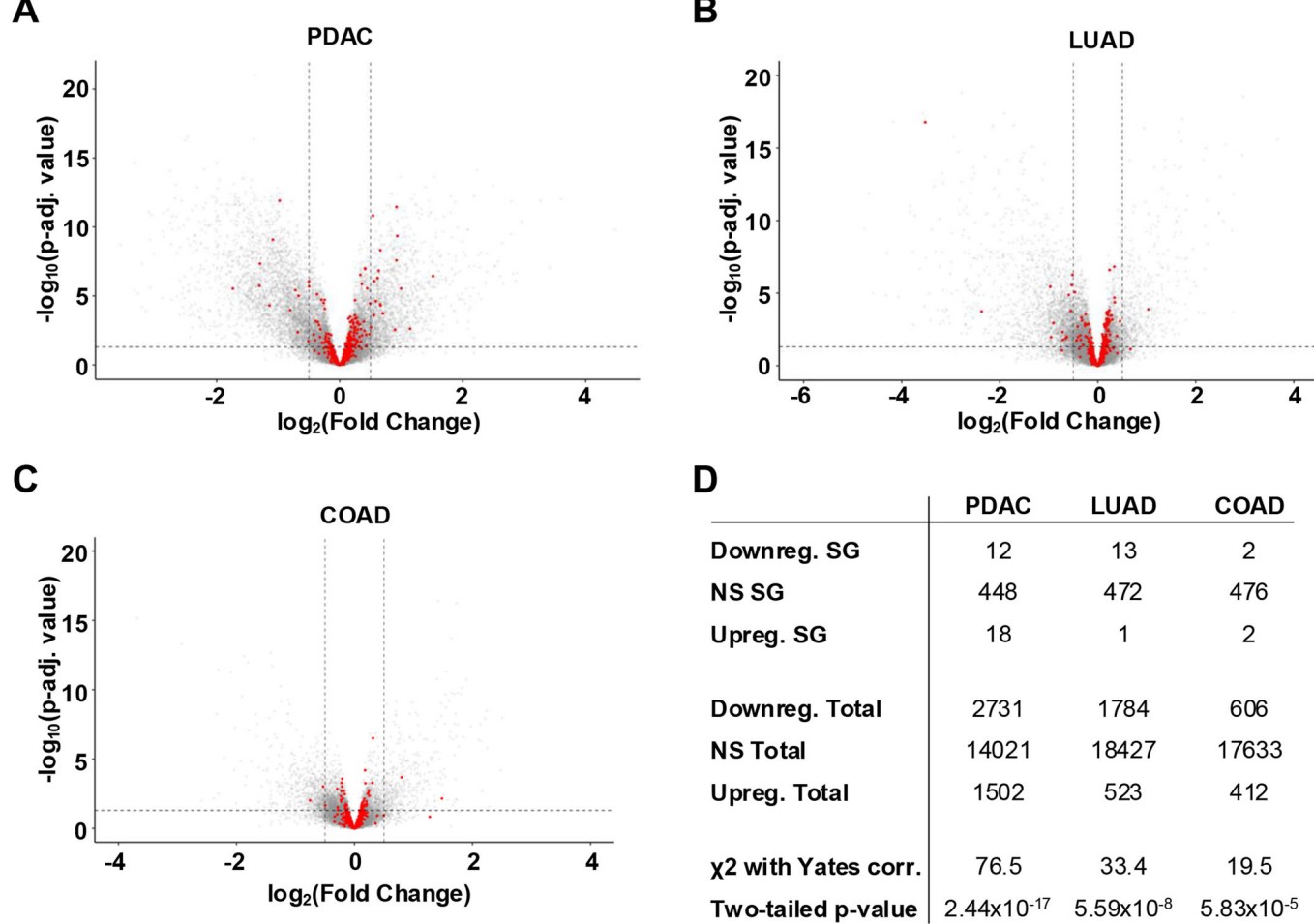

**Figure 4.   KRAS mutations are not associated with transcription of genes encoding SG protein.**

Volcanoplots of DESeq2 results showing gene expression changes in PDAC (A), LUAD (B), and COAD (C). Gene encoding SG proteins (SG score ≥ 4) are shown as red dots. The logarithms of the fold changes of individual genes to base 2 (x-axis) are plotted against the negative logarithm of their p-value to base 10 (y-axis). (D) Table representing the number of downregulated and upregulated genes encoding SG proteins (SG score ≥ 4) of the volcanoplots in (A), (B), and (C). Positive log$_2$(fold change) values represent upregulation and negative values represent downregulation. Chi-square with Yates correction was used to calculate two-tailed p-value. Source data are available online for this figure.

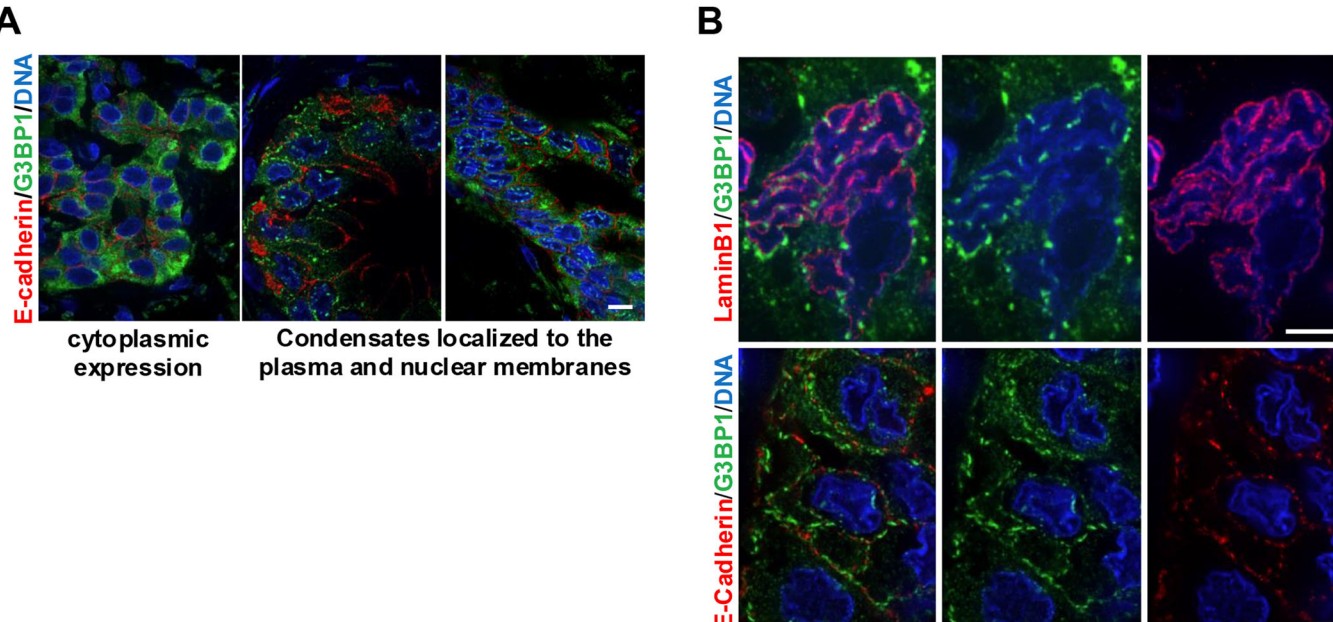

**Figure 5. Cytoplasmic SG are not detected in human PDAC.**

(**A**) Immunolabeling for G3BP1 and E-cadherin (plasma membrane marker) performed on human PDAC sections. Left panel: cytoplasmic expression of G3BP1. Middle and right panels: G3BP1 condensates. Nuclei were counterstained with Hoechst. Scale bar: 10 μm. (**B**) Immunolabeling for G3BP1, LaminB1 (nuclear lamina marker), and E-cadherin performed on human PDAC sections. No correlation was found between the presence of these G3BP1 condensates and obesity. Nuclei were counterstained with Hoechst. Scale bar: 5 μm. The pictures shown are representative of tissues from at least 3 biological replicates. Source data are available online for this figure.

## No evidence supports cytoplasmic SG formation in human PDAC

Besides the presence of SG in animal models of cancer, SG have also been described in human pancreatic cancer specimens (Grabocka and Bar-Sagi, 2016). In addition, a subsequent study claimed that SG are upregulated in obesity-associated PDAC which is dependent on SG for its accelerated growth, compared to PDAC not associated with obesity (Fonteneau et al, 2022). To explore this possibility, we performed G3BP1 labeling on PDAC sections from non-obese and obese patients. We found that, both in non-obese and obese patients, G3BP1 was mainly present in the cytoplasm of human PDAC cells, although in some areas G3BP1 condensates were also present (Fig. 5A). In these areas, colabelling experiments showed that G3BP1 was mainly localized to the inner nuclear membrane and the plasma membrane of PDAC cells, cellular localizations that do not correspond to the presence of G3BP1 in cytoplasmic SG (Fig. 5B). Strikingly we observed that the nuclei with G3BP1 condensates displayed irregular nuclear contours, characteristic of nuclear atypia. Referring to published data (Fonteneau et al, 2022), we noted that G3BP1 labeling was also concentrated in the cell nuclei, which does not fit with cytoplasmic SG, and between the PDAC cells, which may correspond to a location at the cell membranes separating the PDAC cells. Together, these observations do not support that cytoplasmic SG are present in pancreatic tumor cells.

In conclusion, our results do not support that SG are present and impact cancer progression in cancers harboring KRAS mutations, at least that their presence is not universal. To the best of our knowledge, the presence of SG has rarely been described in tumors, essentially in a form of medulloblastoma with a mutation in a protein that promotes SG assembly (Somasekharan et al, 2015; Valentin-Vega et al, 2016). Also, the presence of SG in samples from patients (or mouse models) treated with chemotherapy, regardless of the drug used, has not been described. In this context, we conclude that caution is required before considering SG as therapeutic targets.

## Methods

**Reagents and tools table**

| Reagent/Resource | Reference or Source | Identifier or Catalog Number |
|---|---|---|
| **Experimental Models** | | |
| ElastaseCre^ER mice | Desai et al, 2007 | N/A |
| LSLKras^G12D mice | Hingorani et al, 2003 | N/A |
| HeLa cell line | ATCC | CCL-2 |
| SkChA1 cell line | ATCC | HTB-39 |
| BXPC3 cell line | ATCC | CRL-1687 |
| U2OS cell line | ATCC | HTB-96 |
| HEK293 cell line | ATCC | CRL-1573 |
| EGI-1 cell line | Cellosaurus | CVCL_1193 |
| Panc1 cell line | ATCC | CRL-1469 |
| HPAC cell line | ATCC | CRL-2119 |
| KPE cell line | This study | N/A |

| Reagent/Resource | Reference or Source | Identifier or Catalog Number |
|---|---|---|
| DLD1 cell line | ATCC | CCL-221 |
| MiaPaCa2 cell line | ATCC | CRL-1420 |
| *Escherichia Coli* | Promega | L2005 |
| **Recombinant DNA** | | |
| Ad5CMVCre | University of Iowa, USA | N/A |
| pEGFP | This study | N/A |
| Citrine-KRAS | Chandra et al, 2012 | N/A |
| GFP-KRAS^{G12V} | Fish et al, 2020 | N/A |
| mouse G3BP1 expression vector | ORIGENE | MR207441 |
| **Antibodies** | | |
| Monoclonal Mouse anti-AMYLASE | Santa Cruz | sc-46657 |
| Polyclonal Rabbit anti-CAPRIN1 | ProteinTech | 15112-1-AP |
| Monoclonal Mouse anti-E-CADHERIN | BD Biosciences | 610182 |
| Monoclonal Mouse anti-G3BP1 | BD Biosciences | 611126 |
| Polyclonal Rabbit anti-G3BP1 | Bethyl | A302-033A |
| Polyclonal Rabbit anti-G3BP1 | ProteinTech | 13057-2-AP |
| Polyclonal Rabbit anti-G3BP2 | Abcam | ab86135 |
| Monoclonal Mouse anti-HSC70 | Santa Cruz | sc-7298 |
| Monoclonal Mouse anti-KRAS4B | Sigma-Aldrich | WH0003845M1 |
| Monoclonal Mouse anti-LAMIN B1 | ProteinTech | 66095-1-Ig |
| Monoclonal Mouse Anti-Myc Tag | Cell Signalling Technology | 2276 |
| Polyclonal Rabbit anti-UBAP2L | Bethyl | A300-534A |
| Monoclonal Mouse anti-β-CATENIN | BD Biosciences | 610154 |
| Polyclonal Goat Anti-Mouse IgG1, Alexa Fluor™ 594 | Thermo Fisher Scientific | A21125 |
| Polyclonal Goat Anti-Mouse IgG2a, Alexa Fluor™ 647 | Thermo Fisher Scientific | A21241 |
| Polyclonal Donkey Anti-Rabbit IgG (H + L), Alexa Fluor™ 488 | Thermo Fisher Scientific | A21206 |
| Polyclonal Goat Anti-Rabbit IgG Antibody, HRP linked | Rockland | KCC003 |
| **Chemicals, Enzymes and other reagents** | | |
| Tamoxifen | Sigma-Aldrich | T5648-1G |
| 4-hydroxytamoxifen | Sigma-Aldrich | H7904-25mg |
| Cerulein | Eurogentec | AS-24252 |
| Domitor | Vetoquinol | N/A |
| Nimatek | Dechra | N/A |
| Vetergesic | Ecuphar | N/A |
| Antisedan | Vetoquinol | N/A |
| Paraformaldehyde | Sigma-Aldrich | HT501128 |
| DMEM | Life Technologies | 61965026 |
| RPMI | Life Technologies | 21875034 |
| MEM NEAA | Life Technologies | 11140-035 |
| FBS | Sigma-Aldrich | F9665-500ML |
| Pen. Strep | Life Technologies | 15070063 |
| Sodium Pyruvate | Life Technologies | 11360070 |
| L-Glutamine | Thermo Fisher Scientific | 25030081 |
| Sodium arsenite | Sigma-Aldrich | S7400 |
| 15-d-PGJ2 | Sanbio | 18570-1 |
| DSP (Disuccinimidyl suberate) | CovaChem | 13303-100 |
| Protease inhibitor cocktail | Sigma-Aldrich | 11836153001 |
| Tween-20 | Sigma-Aldrich | P2287 |
| SuperSignal™ West Pico PLUS Chemiluminescent Substrate | Thermo Fisher Scientific | 34577 |
| Polylysine | Sigma-Aldrich | P2636 |
| Triton X-100 | Carl Roth | 3051.3 |
| Bovine serum albumin | Sigma-Aldrich | A906-100G |
| Hoechst | Sigma-Aldrich | B2261 |
| Fluorescence mounting medium | Agilent | S302380-2 |
| HBSS | Life Technologies | 14025-050 |
| Collagenase P | Sigma-Aldrich | 11213865001 |
| DSP (Disuccinimidyl suberate) | CovaChem | 13303-100 |
| DMSO 5% | Sigma-Aldrich | D4540 |
| Tris | MP Biomedicals | 04819638 |
| NaCl | Carl Roth | P029.3 |
| Igepal® | Sigma-Aldrich. | I8896 |
| glycerol | MP Biomedicals | 800688 |
| EDTA | MP Biomedicals | 04800683 |
| Protein A/G Magnetic Beads | Thermo Fisher Scientific | 88802 |
| Laemmli Sample Buffer | Bio-Rad | 1610747 |
| 2-mercaptoethanol | Life Technologies | 31350010 |
| **Software** | | |
| Halo software | Indica Labs | v.3.3.2541 |
| ZEN software | Zeiss | N/A |
| Proteome Discoverer | Thermo Fisher Scientific | v2.5 SP1 |
| TCGAbiolinks R-package | Colaprico et al, 2016 | v2.14.1 |
| GraphPad Prism software | GraphPad Software Inc., San Diego, CA, United States | v9.0.0 |
| **Other** | | |
| Pure Yield™ Plasmid Midiprep system | Promega | A2495 |
| NanoDrop™ One | Thermo Fisher Scientific | N/A |

| Reagent/Resource | Reference or Source | Identifier or Catalog Number |
|---|---|---|
| Bradford assay | Thermo Fisher Scientific | 23200 |
| Polyvinylidene difluoride membranes | Millipore | ISEQ00010 |
| Amersham ImageQuant 800 imaging system | Cytiva | N/A |
| Lab Vision PT Module | Thermo Fisher Scientific | N/A |
| reversed-phase pre-column | Thermo Fisher Scientific | Acclaim PepMap 100 |
| reversed-phase analytical column | Thermo Fisher Scientific | Acclaim PepMap RSLC |
| Orbitrap Fusion Lumos tribrid mass spectrometer | Thermo Fisher Scientific | N/A |
| ELISA kit | Enzo Life Sciences | ADI-900-023 |

## Mouse model and treatments

All procedures described in this study were performed with the approval of the animal welfare committee of the UCLouvain Health Sciences Sector (Brussels, Belgium; ethic number: 2021/UCL/MD/054). ElaK and ElaKP mice (Desai et al, 2007; Hingorani et al, 2003; Hingorani et al, 2005) were maintained in a CD1-enriched background. These mouse models allow the expression of mutated $Kras^{G12D}$ and $p53^{R172H}$ alleles specifically in acinar cells. Six- to eight-week-old ElaK and ElaKP mice were treated with 100 µl tamoxifen (T5648-1G, Sigma-Aldrich. 3 mg in 100 µl of corn oil) by oral gavage, and 100 µl 4-hydroxytamoxifen (H7904-25mg, Sigma-Aldrich. 30 µg in 100 µL of corn oil) by subcutaneous injection. Three sets of tamoxifen and 4-hydroxytamoxifen treatments, each separated by 48 h, allowed the recombination of the LSL cassettes and the expression of $Kras^{G12D}$ and $p53^{R172H}$ from their respective endogenous locus. To induce acute pancreatitis, tamoxifen-treated ElaK and ElaKP mice received 7 hourly intraperitoneal injections of cerulein (AS-24252, Eurogentec. 100–150 µl in PBS, pH 7.4, 125 µg/kg), every other day, for 5 days. ElaKP mice were subjected to chronic inflammation by additional cerulein treatment consisting of one injection a day, three times a week, during four weeks. To allow PDAC development, the ElaKP mice were then kept alive until they reached 32 weeks.

Orthotopic tumor development was carried out by injection of ElaKP-derived cells in the pancreas of 6-week-old CD1 mice (Qiu and Su, 2013). Before surgery, mice were anesthetized by intraperitoneal administration of 10 µl PBS/10% Domitor (Vetoquinol)/7.5% Nimatek (Dechra) per g body weight. A total of $5 \times 10^5$ ElaKP-derived tumor cells resuspended in 50 µl PBS were injected in the head of the pancreas. Postoperative awaking was mediated by subcutaneous injection of Vetergesic (Ecuphar. 16.7 µL/g) and Antisedan (Vetoquinol. 10 µL/g). After orthotopic implantation, mice were regularly monitored by visual observation for tumor growth and general signs of morbidity. Twenty-one days after implantation, the mice were sacrificed.

To induce lung tumors, adenoviral vectors expressing the Cre recombinase protein under the control of the CMV promoter (Ad5CMVCre, University of Iowa, USA), were introduced via intratracheal instillation in $LSLKras^{G12D}$ $LSLRosa^{YFP}$ mice (DuPage et al, 2009).

Mice were sacrificed by cervical dislocation and pancreas or lungs were collected for subsequent analyses.

## Heat shock

For heat shock, mice were anesthetized and their pancreas was carefully taken out of the abdominal cavity. The exposed pancreas was then incubated in PBS at 37 °C or 43 °C, for 30 min. After sacrifice, the pancreas was fixed in 4% paraformaldehyde (HT501128, Sigma-Aldrich) at 4 °C for 4 h, before embedding in paraffin.

## Cell lines

Cells were cultured in a humidified 5% $CO_2$ incubator at 37 °C. ElaKP cells were derived from a tumor induced in ElaKP mice. HeLa, SkChA1, BXPC3, U2OS, and HEK293 cell lines do not have KRAS mutations, while EGI-1, Panc1, HPAC, KPE, DLD1, and MiaPaCa2 cell lines bear a KRAS mutation.

## Arsenite and 15-d-PGJ2 treatments

Formation of SG was induced by incubating cells with 0.1 mM sodium arsenite (Sigma-Aldrich, S7400) for 30 min. Cells were treated with 50 µM 15-d-PGJ2 (Sanbio, 18570-1) for 60 min.

## Transfection

30,000 cells were seeded in a 24-well plate for SG quantification or $10^6$ cells in 10-cm diameter dish for validation of transfection efficiency by Western blot. Calcium phosphate transfection (Chen 2012) was performed with 0.21 µg DNA/cm². The plasmids used for transfection were pEGFP, Citrine-KRAS (Chandra et al, 2012) and $GFP-KRAS^{G12V}$ (Fish et al, 2020). To validate the anti-G3BP1 antibodies, $1.5 \times 10^5$ HeLa cells were seeded in 10-cm diameter dish and transfected with 1 µg of mouse G3BP1 expression vector (MR207441, ORIGENE) as previously described (Episkopou et al, 2019). Plasmids were amplified in competent *Escherichia coli* (L2005, Promega) following the manufacturer's instructions. The plasmids were purified using Pure Yield™ Plasmid Midiprep system (A2495, Promega) and their concentration were determined using NanoDrop™ One (Thermo Fisher Scientific).

## SG quantification

SG quantifications were performed on 10 randomly acquired images at 40× magnification per replicate for each cell line. The Halo software (Indica Labs, v.3.3.2541) was employed for SG quantification. Using an algorithm (Indica Labs – Object colocalization FL v1.0), a threshold was set to distinguish SG from background noise, and the software automatically detected and quantified individual granules based on size, intensity, and shape parameters. Using another algorithm (Indica Labs – Area quantification FL v2.1.7), a second threshold was set to determine the cytoplasmic area of the cells. The SG index was calculated by dividing the total surface occupied by SG by the total surface of the cells for each image (multiplied by 100). This calculation was performed for every image, and the average value was obtained by combining the results from all images.

## Western blotting

$10^6$ cells were seeded in 10-cm diameter dish. Cells were lysed by vortexing and repetitive pipetting in a buffer composed of 50 mM Tris-Cl, 150 mM sodium chloride, 0.25% sodium deoxycholate, 1% NP-40, 1 mM sodium orthovanadate, 2% sodium dodecyl sulfate (SDS), containing a protease inhibitor cocktail (11836153001, Sigma-Aldrich). Lysates were maintained on ice during the procedure. Then, cell debris were pelleted by centrifugation ($14,000 \times g$, 10 min, 4 °C). Proteins were quantified using a Bradford protein assay kit (23200, Thermo Fisher Scientific). Lysates containing 50 µg total proteins were separated on 12.5% SDS polyacrylamide gels. Polyvinylidene difluoride membranes (ISEQ00010, Millipore) were blocked with a solution of 5% low-fat milk diluted in Tris-buffered saline (TBS)/0.1% Tween-20 (P2287, Sigma-Aldrich) for 1 h at room temperature (RT). Membranes were incubated overnight at 4 °C with primary antibodies against KRAS4B (WH0003845M1, Sigma-Aldrich. 1 µg/ml), anti-G3BP1 (13057-2-AP, ProteinTech. 0.3 µg/ml), anti-G3BP1 (611126, BD Biosciences. 0.25 µg/ml), anti-G3BP1 (A302-033A, Bethyl. 0.04 µg/ml), anti-Myc Tag (2276, Cell Signalling Technology. 0.08 µg/ml), or HSC70 (sc-7298, Santa Cruz. 40 ng/ml) used as loading control, diluted in blocking buffer. Then, membranes were washed with TBS/0.1% Tween-20 and incubated with secondary antibodies for 1 h at RT. After incubation, membranes were washed with TBS/0.1% Tween-20 and signals were revealed using SuperSignal™ West Pico PLUS Chemiluminescent Substrate (34577, Thermo Fisher Scientific). Pictures were taken with the Amersham ImageQuant 800 imaging system (Cytiva).

## Human pancreas specimens

The use of human tissue with histologically confirmed pancreatic ductal adenocarcinoma and without neoadjuvant treatment was approved by the Committee of Medical Ethics—Erasme Hospital (P2021/382). BMI data were obtained for each patient.

## Immunofluorescence

60,000 cells were seeded on a coated-polylysin (P2636, Sigma-Aldrich. 50 µg/ml in PBS, 25 min, 37 °C) coverslip in 24-well plate. One day later, cells were fixed with 4% paraformaldehyde at 4 °C. Six-µm paraffin-embedded tissue sections were deparaffinized and antigen retrieval was performed by heating the slides for 20 min in Tris-EDTA buffer (pH 9) using Lab Vision PT Module (Thermo Fisher Scientific). Sections and coverslips were rinsed 5 min in PBS, and permeabilized in PBS, 0.3% Triton X-100 (3051.3, Carl Roth) for 5 min at RT. After permeabilization, samples were blocked in PBS/3% low-fat milk/10% bovine serum albumin (A906-100G, Sigma-Aldrich)/0.3% Triton X-100 (blocking buffer) for 45 min at RT. Primary antibodies were diluted in blocking buffer and incubated with the samples overnight at 4 °C. Primary antibodies are listed in the Reagents and Tools table. Then, samples were washed with 0.1% Triton X-100 in PBS. Secondary antibodies were diluted in PBS, 10% bovine serum albumin, 0.3% Triton X-100, and incubated for 1 h at 37 °C. Nuclei were counterstained with Hoechst (B2261, Sigma-Aldrich. 3.2 µM). Then, samples were washed with 0.1% Triton X-100 in PBS and the slides were mounted in fluorescence mounting medium (S302380-2, Agilent). Imaging was

performed with Cell Observer Spinning Disk Confocal Microscope using the ZEN software. Laser power and exposure times were similar for images from each datasets.

## Immunoprecipitation and mass spectrometry

Minced pancreas was washed in HBSS (14025-050, Life Technologies) and digested in Collagenase P (11213865001, Sigma-Aldrich. 0.2 mg/ml in HBSS) during 15 min at 37 °C at 225 rpm. Enzymatic activity was inhibited with cold 2.5% FBS. Tissue fragments were washed twice with PBS at 4 °C and crosslinked by incubation at RT in freshly prepared DSP (13303-100, CovaChem. 2 mM in PBS/DMSO 5% (D4540, Sigma-Aldrich)). Cross-linking reaction was then quenched by adding Tris (04819638, MP Biomedicals. 20 mM, pH 7.5) for 15 min. Proteins were extracted from centrifuged cell pellet using RIPA lysis buffer (10 µl/mg of tissue) composed of Tris (50 mM, pH 7.5), NaCl (P029.3, Carl Roth. 100 mM), Igepal® (I8896, Sigma-Aldrich. 0.5%), glycerol (800688, MP Biomedicals. 10%), EDTA (04800683, MP Biomedicals. 125 µM), $MgCl_2$ (1058330250, Sigma-Aldrich. 0.5 mM) and protease inhibitor. The lysate was clarified by centrifugation. Protein concentration was determined using the Bradford protein assay. Pre-washed protein A/G magnetic beads (88802, Thermo Fisher Scientific. 50 µl per sample) were coupled with 10 µg of the specific antibody against the target protein in a total volume of 200 µl RIPA buffer. IgG anti-rabbit antibody (KCC003, Rockland) was used as isotype control antibody. The washed antibody-conjugated beads were then incubated with the pre-cleared cell lysate overnight at 4 °C, with gentle agitation. Following immunoprecipitation, the beads were washed extensively with RIPA buffer to remove nonspecifically bound proteins. The immunoprecipitated proteins were eluted from the beads using 4x Laemmli Sample Buffer (1610747, Bio-Rad) with 2-mercaptoethanol (31350010, Life Technologies) supplemented with RIPA buffer to a final volume of 50 µl. Immunoprecipitated products were subjected to Western Blot to verify immunoprecipitation of the protein of interest. 50 µg of protein of whole cell lysate was used as input. In parallel, proteins were separated in a 10% acrylamide gel with a short 15-min run at 120 V. Protein bands were visualized by colloidal Coomassie Blue staining and in-gel digested with trypsin. Peptides were extracted with 0.1% TFA in 65% acetonitrile (ACN) and dried in a SpeedVac vacuum concentrator.

Peptides were dissolved in solvent A (0.1% TFA in 2% ACN), directly loaded onto reversed-phase pre-column (Acclaim PepMap 100, Thermo Fisher Scientific) and eluted in backflush mode. Peptide separation was performed using a reversed-phase analytical column (Acclaim PepMap RSLC, 0.075 × 250 mm, Thermo Fisher Scientific) with a linear gradient of 4–27.5% solvent B (0.1% FA in 98% ACN) for 40 min, 27.5–50% solvent B for 20 min, 50–95% solvent B for 10 min, and holding at 95% for the last 10 min at a constant flow rate of 300 nl/min on an Ultimate 3000 RSLC system. The peptides were analyzed by an Orbitrap Fusion Lumos tribrid mass spectrometer (Thermo Fisher Scientific). The peptides were subjected to NSI source followed by tandem mass spectrometry (MS/MS) in Fusion Lumos coupled online to the nano-LC. Intact peptides were detected in the Orbitrap at a resolution of 120,000. Peptides were selected for MS/MS using HCD setting at 30, and ion fragments were detected in the Orbitrap at a resolution of 30,000. A data-dependent procedure

that alternated between one MS scan followed by MS/MS scans was applied for 3 s for ions above a threshold ion count of $2 \times 10^4$ in the MS survey scan with 40.0 s dynamic exclusion. The electrospray voltage applied was 2.1 kV. MS1 spectra were obtained with an AGC target of $4 \times 10^5$ ions and a maximum injection time of 50 ms, and MS2 spectra were acquired with an AGC target of $5 \times 10^4$ ions and a maximum injection set to dynamic. For MS scans, the $m/z$ scan range was 375 to 1800. The resulting MS/MS data was processed using Sequest HT search engine within Proteome Discoverer 2.5 SP1 against a *Mus Musculus* protein database obtained from Uniprot. Trypsin was specified as cleavage enzyme allowing up to 2 missed cleavages, 4 modifications per peptide and up to 5 charges. Mass error was set to 10 ppm for precursor ions and 0.1 Da for fragment ions. Oxidation on Met ($+15.995$ Da), conversion of Gln ($-17.027$ Da) or Glu ($-18.011$ Da) to pyro-Glu at the N-term peptide were considered as variable modifications. False discovery rate (FDR) was assessed using Percolator and thresholds for protein, peptide, and modification site were specified at 1%. For abundance comparison, abundance ratios were calculated by Label Free Quantification (LFQ) of the precursor intensities within Proteome Discoverer 2.5 SP1. Proteins were considered as immunoprecipitated with G3BP1 or with G3BP2 if abundance ratio ≥2, adjusted $p$-value ≤ 0.05, Sum PEP score ≥3 and FDR discovery = high.

### ELISA

15-d-PGJ2 levels were measured using an ELISA kit (ADI-900-023, Enzo Life Sciences) following the manufacturer's instructions. The measurements were carried out in conditioned medium, 72 h after seeding 30,000 cells in a 24-well plate.

### Analysis of human patient data

Raw STAR counts, transcriptomic data normalized in transcripts per kilobase million (TPM) and the Single Nucleotide Variant (SNV) datasets for pancreatic ductal adenocarcinoma (PDAC), lung adenocarcinoma (LUAD) and colon adenocarcinoma (COAD) were downloaded from The Cancer Genome Atlas (TCGA) consortium (Grossman et al, 2016), using TCGAbiolinks v2.14.1 R-package (Colaprico et al, 2016). Only unique primary (01A) tumor samples with available RNA-seq and SNV data were included in the analysis. Pancreatic neuroendocrine carcinoma samples ($n = 8$) were excluded for the PDAC analysis. The tumor samples were categorized on the possible presence of a *KRAS* missense mutation, either in wild-type (*KRAS* WT) or in mutant (*KRAS* MUT). For PDAC: KRAS WT = 58; KRAS MUT = 99. For LUAD: KRAS WT = 364; KRAS MUT = 138. For COAD: KRAS WT = 241; KRAS MUT = 180.

### Statistical analysis

All graphical and statistical analyses were conducted using GraphPad Prism 9.0.0 software (GraphPad Software Inc., San Diego, CA, United States). Data are presented as mean ± standard error of the mean (SEM). For single comparisons between two experimental groups, an unpaired Student's t-test was performed. To identify significant differences between multiple groups, data were subjected to a one-way or two-way ANOVA with Tukey's correction for multiple comparisons. Shapiro–Wilk normality tests

and QQ plots were performed to assess the normality of the residuals of the data. Homogeneity of variances was verified with Fisher's test. For all statistical analyses, the level of significance was set at $p < 0.05$. Statistical significance is indicated on the graphs with the following notation: $*p < 0.05$; $**p < 0.01$; $***p < 0.001$.

DESeq2 analyses were used for read counting and differential gene expression between the KRAS WT and KRAS MUT samples and to generate volcanoplots (Love et al, 2014). Differentially expressed genes are highlighted with significant adjusted $p$-values < 0.05 and $\log_2$ Fold Change >0.5 or <−0.5.

## Data availability

The mass spectrometry proteomics datasets produced in this study have been deposited to the ProteomeXchange Consortium via the PRIDE repository with the dataset identifier PXD053882.

The source data of this paper are collected in the following database record: biostudies:S-SCDT-10_1038-S44319-024-00284-6.

## Peer review information

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

## Acknowledgements

We thank members of the laboratory of Patrick Jacquemin and Frederic Lemaigre for help and discussions, Frederic Lemaigre for comments, Malak Haidar for KPE cells, Guido Bommer's laboratory for DLD1 and MiaPaCa2 cells, Jean-Nicolas Lodewyckx and Hajar Dahou for mouse genotyping, and Mourad El Kaddouri and Bruno Maricq for their help in mouse care. We also would like to thank Craig B. Wilen for providing mouse G3BP1 expression (ORIGENE, MR207441). This work was supported by grants from FRS-FNRS (J.0096.21 and J.0097.23 to P Jacquemin) and Télévie (7.8503.20 to P Jacquemin). Maxime Libert held Télévie fellowships (7.4553.19 and 7.6524.21). P Jacquemin is Research Director at FRS-FNRS, Belgium.

## Author contributions

**Maxime Libert**: Conceptualization; Data curation; Validation; Investigation; Methodology; Writing—review and editing. **Sophie Quiquempoix**: Conceptualization; Data curation; Validation; Investigation; Methodology; Writing—review and editing. **Jean S Fain**: Conceptualization; Data curation; Validation; Investigation; Methodology; Writing—review and editing. **Sébastien Pyr dit Ruys**: Validation; Methodology; Writing—review and editing. **Malak Haidar**: Methodology; Writing—review and editing. **Margaux Wulleman**: Methodology; Writing—review and editing. **Gaëtan Herinckx**: Validation; Methodology. **Didier Vertommen**: Validation; Methodology; Writing—review and editing. **Christelle Bouchart**: Methodology; Writing—review and editing. **Tatjana Arsenijevic**: Methodology; Writing—review and editing. **Jean-Luc Van Laethem**: Methodology; Writing—review and editing. **Patrick Jacquemin**: Conceptualization; Data curation; Supervision; Funding acquisition; Validation; Investigation; Methodology; Writing —original draft; Writing—review and editing.

Source data underlying figure panels in this paper may have individual authorship assigned. Where available, figure panel/source data authorship is listed in the following database record: biostudies:S-SCDT-10_1038-S44319-024-00284-6.

## Disclosure and competing interests statement

The authors declare no competing interests.

# Expanded View Figures

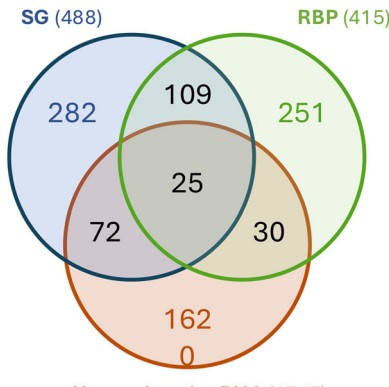

**Figure EV1.  Several mRNA coding for RBP and found in SG are present among the mRNA upregulated during pancreatic inflammation.**

Venn diagram developed from transcriptomic data previously collected by our team (Assi et al, 2021), the RBP databank (http://rbpdb.ccbr.utoronto.ca/), and the SG databank (https://rnagranuledb.lunenfeld.ca/).

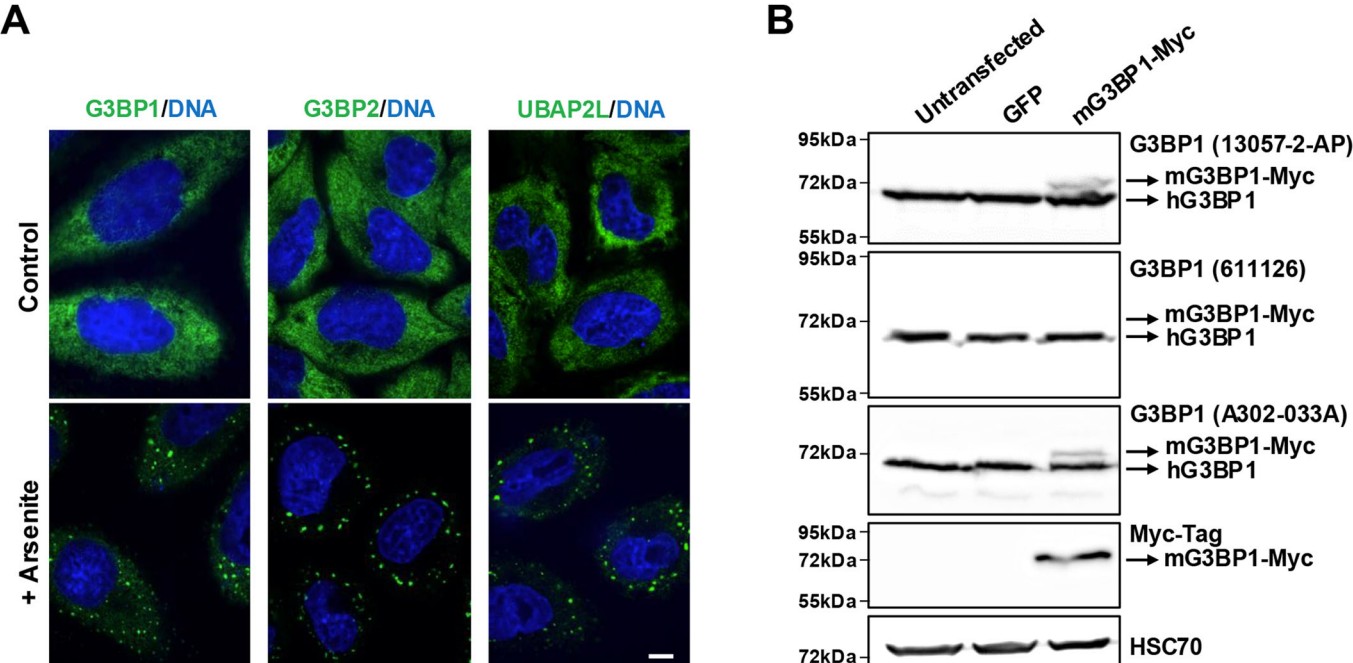

**Figure EV2. Selected antibodies specifically recognize SG proteins.**

(A) Immunolabelling performed with G3BP1 (13057-2-AP), G3BP2, and UBAP2L antibodies on untreated (top panels) or arsenite-treated (bottom panels) Hela cells. Scale bar: 10 µm. (B) Western blotting performed on protein extracts from untransfected, GFP-transfected, and mouse (m) G3BP1-transfected HeLa cells with G3BP1 (13057-2-AP), G3BP1 (611126), G3BP1 (A302-033A), and Myc-Tag antibodies. Human (h) G3BP1 was detected by the 3 G3BP1 antibodies, whereas mG3BP1 was recognized by G3BP1 (13057-2-AP) and G3BP1 (A302-033A) antibodies, but not by G3BP1 (611126) antibody. The pictures shown are representative of tissues from at least 3 biological replicates.

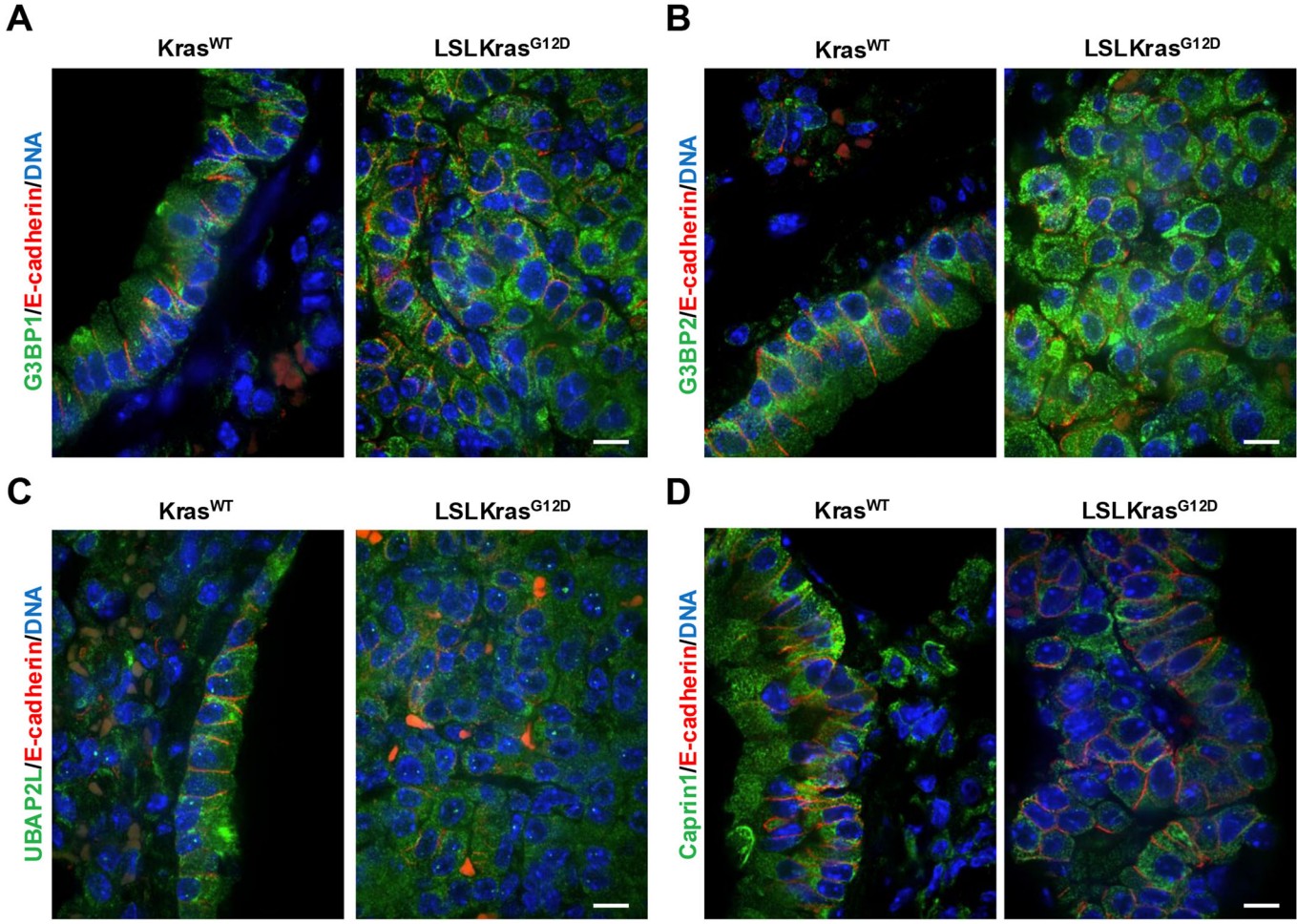

**Figure EV3. SG are not observed in lung adenomas.**

Immunolabeling performed with E-cadherin and G3BP1 (A), G3BP2 (B), UBAP2L (C), and Caprin1 (D) on lung sections of Kras[WT] and LSLKras[G12D] mice infected by intratracheal instillation with Cre adenovirus. Bronchial epithelium is shown for Kras[WT] lung sections while adenoma is exemplified for LSLKras[G12D] lung sections. Nuclei were counterstained with Hoechst. Scale bar: 10 μm. The pictures shown are representative of tissues from at least 3 biological replicates.

