## [Peer Review File · EMBO Reports]

Stress granules are not present in Kras mutant cancers and do not control tumor growth

Maxime Libert, Sophie Quiquempoix, Jean Fain, Sébastien Pyr dit Ruys, Malak Haidar, Margaux Wulleman, Gaëtan Herinckx, Didier Vertommen, Christelle Bouchart, Tatjana Arsenijevic, Jean-Luc Van Laethem, and Patrick Jacquemin

Corresponding author(s): Patrick Jacquemin (patrick.jacquemin@uclouvain.be)

Review Timeline:

Submission Received:	3rd Feb 24
Editorial Decision:	4th Mar 24
Revision Received:	18th Jul 24
Editorial Decision:	21st Aug 24
Revision Received:	25th Sep 24
Accepted:	30th Sep 24

Editor: Achim Breiling

Transaction Report:

Dear Dr. Jacquemin,

Thank you for the submission of your manuscript to EMBO reports. I have now received the reports from the three referees that were asked to evaluate your study, which can be found at the end of this email.

As you will see, the referees think that the findings are of interest. Although referee #1 indicates that the study remains rather descriptive and of moderate advance, referee #3 states that the manuscript will be an important contribution to the fields of stress granules and cancer. Both referees do not indicate any requests for revision.

In contrast, referee #2 has several comments, concerns, and suggestions, indicating that a major revision of the manuscript is necessary to allow publication of the study in EMBO reports. As the report is below, and all the concerns need to be addressed, I will not detail them here.

Given the constructive referee comments, I would like to invite you to revise your manuscript with the understanding that the concerns of referee #2 must be addressed in the revised manuscript or in a detailed point-by-point response. Acceptance of your manuscript will depend on a positive outcome of a second round of review. It is EMBO reports policy to allow a single round of revision only and acceptance of the manuscript will therefore depend on the completeness of your responses included in the next, final version of the manuscript.

- 1) a .docx formatted version of the final manuscript text (including legends for main figures, EV figures and tables), but without the figures included. Figure legends should be compiled at the end of the manuscript text.
- 2) individual production quality figure files as .eps, .tif, .jpg (one file per figure), of main figures and EV figures. Please upload these as separate, individual files upon re-submission.

- 4) a complete author checklist, which you can download from our author guidelines

(<https://www.embopress.org/page/journal/14693178/authorguide>). Please insert page numbers in the checklist to indicate where the requested information can be found in the manuscript. The completed author checklist will also be part of the RPF.

Please also follow our guidelines for the use of living organisms, and the respective reporting guidelines:
<http://www.embopress.org/page/journal/14693178/authorguide#livingorganisms>

5) that primary datasets produced in this study (e.g. RNA-seq, ChIP-seq, structural and array data) are deposited in an appropriate public database. If no primary datasets have been deposited, please also state this in a dedicated section (e.g. 'No primary datasets have been generated and deposited'), see below.

The accession numbers and database should be listed in a formal "Data Availability" section (placed after Materials & Methods) that follows the model below. This is now mandatory (like the COI statement). Please note that the Data Availability Section is restricted to new primary data that are part of this study. This section is mandatory. As indicated above, if no primary datasets have been deposited, please state this in this section

Data availability

8) Regarding data quantification and statistics, please make sure that the number "n" for how many independent experiments were performed, their nature (biological versus technical replicates), the bars and error bars (e.g. SEM, SD) and the test used to calculate p-values is indicated in the respective figure legends (also for EV figures and all those in an Appendix). Please also check that all the p-values are explained in the legend, and that these fit to those shown in the figure. Please provide statistical testing where applicable. Please avoid the phrase 'independent experiment', but clearly state if these were biological or technical replicates. Please also indicate (e.g. with n.s.) if testing was performed, but the differences are not significant. In case n=2, please show the data as separate datapoints without error bars and statistics. See also:
<http://www.embopress.org/page/journal/14693178/authorguide#statisticalanalysis>

9) Please add scale bars of similar style and thickness to microscopic images, using clearly visible black or white bars (depending on the background). Please place these in the lower right corner of the images themselves. Please do not write on or near the bars in the image but define the size in the respective figure legend.

10) Please also note our reference format:
<http://www.embopress.org/page/journal/14693178/authorguide#referencesformat>

12) We now use CRediT to specify the contributions of each author in the journal submission system. CRediT replaces the author contribution section. Please use the free text box to provide more detailed descriptions and do not provide your final

manuscript text file with an author contributions section. See also our guide to authors:
<https://www.embopress.org/page/journal/14693178/authorguide#authorshipguidelines>

13) We would encourage you to use 'Structured Methods', our new Materials and Methods format. According to this format, the Materials and Methods section should include a Reagents and Tools Table (listing key reagents, experimental models, software and relevant equipment and including their sources and relevant identifiers), uploaded as separate file, followed by a Methods and Protocols section in which we encourage the authors to describe their methods using a step-by-step protocol format with bullet points, to facilitate the adoption of the methodologies across labs. More information on how to adhere to this format as well as downloadable templates (.doc or .xls) for the Reagents and Tools Table can be found in our author guidelines (section 'Structured Methods'):

If you provide a Reagents and Tools Table, please remove the tables from the methods section.

14) Please provide the abstract written in present tense and order the manuscript sections like this, using these names:
Title page - Abstract - Keywords - Introduction - Results & Discussion - Materials and Methods - Data availability section - Acknowledgements - Disclosure and Competing Interests Statement - References - Figure legends - Expanded View Figure legends

Finally, please note that all corresponding authors are required to supply an ORCID ID for their name upon submission of a revised manuscript. Please find instructions on how to link the ORCID ID to the account in our manuscript tracking system in our Author guidelines: <http://www.embopress.org/page/journal/14693178/authorguide#authorshipguidelines>

I look forward to seeing a revised version of your manuscript when it is ready. Please let me know if you have questions or comments regarding the revision.

Yours sincerely,

Referee #1:

This is a descriptive study with limited translational value, and mostly representing the negative outcomes of the study without leading to a potential role of the components of the SG in PDAC biology (or SG as this is not completely rule out by the data included in this version of the manuscript). The manuscript is well written and experiments are well executed. However, without the aforementioned mechanistic evidence, the impact of the manuscript is low to moderate. I am sorry that I can't be more positive about the study.

Referee #2:

Stress granules (SGs) are associated with KRAS mutant pancreatic cancer and they have functional roles, in providing resistance to chemotherapeutic agents(oxaliplatin). The presence of SGs in pancreatic cancer is mainly supported by two

studies (Grabocka, E. & Bar-Sagi, D. Cell, 2016)) and Fonteneau, G.Graboka, E, Cancer Discovery, 2022).

In this manuscript, Libert et al, have challenged the conclusions of these two papers and claim stress granules are not formed during tumorigenesis of KRAS-driven cancers.

Stress granules are a form of biomolecular condensates that assemble many proteins and RNA via liquid-liquid phase separation(LLPS). LLPS field of research generated tremendous interest over the past few years and is considered to be a promising drug target. However, sample processing and imaging artifacts that create false-positive LLPS, are major challenges to ascertain biomolecular condensates in specific contexts. Hence rigorous validation is required to show the functional importance of condensates in biological processes.

Although Libert et al have used a variety of models and approaches to show SGs do not form in KRAS mutant cancers, their experimental evidence is more observational/ phenotypic and is not functional. Hence the arguments are not sufficient to invalidate the study published by Bar-sagi and Grabocka. For example, the authors have not used the same cell lines that were used in previous studies. There are not many positive or negative controls used in

the current manuscript. The quantification data are not supported by corresponding microscopy images. As the figure legends and text are not detailed, it is difficult to compare and draw a concrete conclusion. Importantly they have not provided any functional evidence that SGs are not required for tumor growth. The authors have not discussed why their data are different from the other two studies. Taken all together, probably it is appropriate to state that, the presence of stress granules in KRAS mutant cancers "is not universal".

Here are some concerns about the conclusions drawn in the current manuscript.

Mutant KRAS does not exert cell non-autonomous control on SG: The authors do not explain which mut/wt KRAS cell lines are used in their study so that the results can be compared to Grabocka et al. While Grabocka et al have used Hela cells to express KRAS ectopically, Libert et al have used 293T cells to study the formation of SGs by KRAS. Grabocka et al show 5-fold difference in SG formation between mut and wt cells, and Libert et al found only a 1.5-fold difference. Hence it suggests that KRAS can induce SG to a different extent depending on the cell line context.

The authors did not find 15-d-PGJ2 induced SG as reported previously. Using a larger data set they could not establish anti-correlation between PTGS2 and HPGD expression in KRAS mutant cancers as reported previously by Grabocka et al. Fonteneau /Grabocka have shown that IGF1, rather than prostaglandin induces SG in obesity-induced KRAS mutant Pancreatic cancer model. These observations can be interpreted as "Prostaglandin mediated SG formation context-dependent". Hence Libert et al should use multiple cell lines including the one shown by Grabocka and Bar-sagi, to support their claim that prostaglandin is not involved in SG formation.

Non-cytoplasmic stress granules. In the current study, the authors have compared obese and non-obese patients and show that the SGs are present on nuclear and plasma membranes in human samples. The authors also point out that Fonteneau et al show nuclear SGs in their published study. They argue that SGs should be cytoplasmic, not nuclear. However, these observations don't invalidate the presence of SGs in human patient samples. Nuclear stress granules have been reported in several publications. Hence it is not clear if the authors conclude that the SGs shown by them or by Fonteneau et al are artifacts or irrelevant. The authors should explain elaborately.

Antibody issue: The authors claim that the human G3BP1 antibody (611126) doesn't recognize mouse isoform(Fig EV2B). As that antibody was used by Grabocka and Bar-sagi, cell, 2016 to prove the presence of SGs in mouse tumors, they question the validity of the data presented by Grabocka et al. However, several points need to be taken into consideration in this context 1) It seems like the exogenous mouse G3BP1 is expressed at a very low level (verified by other antibodies used in Fig EV2B). It may be possible or G3BP1 (611126) antibody gives a weak signal and is not apparent due to image processing with high contrast (Fig EV2B). Multiple exposures may have helped. 2) Some antibodies may not be suitable for immunoblot but can show a signal in IF or IHC. 3) In Fig EV2B, the authors have shown that G3BP1 antibody (A302- 033A) recognizes mouse G3BP1. This antibody was also used by Fonteneau et al, to show the existence of mouse PanINs (see methods). Hence the argument is not strong enough to refute the previous findings.

Referee #3:

I think that this work is very important and timely. The results and conclusions of this work are clear, and based on the expanded repertoire of models, omics- datasets and validated antibodies. This will be important paper for the field of SG and cancer, which will trigger further re-visit of some SG-related paradigms in cancer. The SG biology and pathobiology area needs such re-evaluation. I support this work for publication.

Referee #2

General comments:

The authors have not used the same cell lines that were used in previous studies. There are not many positive or negative controls used in the current manuscript. The quantification data are not supported by corresponding microscopy images. The authors have not discussed why their data are different from the other two studies. Taken all together, probably it is appropriate to state that the presence of stress granules in KRAS mutant cancers "is not universal". Below are some concerns about the conclusions drawn in the current manuscript. Some general comments being included below as specific comments, we only respond here to the points which are only addressed here.

About quantification data not supported by corresponding microscopy images, we have now added microscopy images to Fig 3A.

Regarding the experiments which show divergent results between the two studies, we believe that they fall into 3 categories:

- experiments in which the final readout is the calculation of the stress index. We think that we probably have the same labeling on our cells and that the differences come from image processing, generating different stress index. We have examined the steps of our processing very carefully and are completely confident that our processing is correct.
- experiments exploring human databanks. For us, the discrepancies come from the number of patients included in the respective analyses. We included many more patients (which was a very easy and obvious thing given the large number of cases found in these banks), which explains the differences between us.
- experiments showing G3BP1 immunolabeling on PDAC samples, both in mice and in humans. In mice, the difference is due to the fact that an antibody that does not recognize the murine form of G3BP1 was used in Grabocka and Bar-Sagi. For humans, we think our results are the same but the interpretation in Grabocka and Bar-Sagi was different (see below for details).

In the original version of our manuscript, as suggested by the referee, we had already highlighted the discrepancies that could be explained in a rational way (inclusion of a small number of patients and use of an antibody incapable of recognizing mouse G3BP1 in their study), but we preferred not to put forward explanations linked to personal interpretations. We have maintained this approach for the revised version.

Specific concerns:

1. Mutant KRAS does not exert cell non-autonomous control on SG.

a) The authors do not explain which mut/wt KRAS cell lines are used in their study so that the results can be compared to Grabocka et al.

The cell lines that we used were described in Materials & Methods (pages 15 and 16 of the original manuscript). We recognize that this information was difficult to find, and accordingly, we have modified Figure 3A to show the names of the cell lines used in a format which is now close to that presented by Grabocka and Bar-Sagi.

Initially, among the 5 wild-type KRAS and 4 mutant KRAS lines that we tested, one mutant line (Panc1) was common between the two studies. The reason why we did not consider it important to use the same lines as Grabocka and Bar-Sagi is explained by the fact that the stress index found by Grabocka is very similar among the 6 mutant Kras lines that they tested, which is also true for the 4 wild-type KRAS lines used, and above all, that this index is

10 times higher in mutant KRAS cells. In this context, Grabocka and Bar-Sagi implicitly indicate that their conclusion is not restricted to the cell lines they tested, but applies to any cell line (following their KRAS status).

However, to address the referee's comment, we have now performed the experiment on two additional mutant KRAS cell lines used by Grabocka and Bar-Sagi (DLD1 and MiaPaCa2). This result is shown in the new Fig 3A. Our conclusions remain unchanged.

b) While Grabocka et al have used HeLa cells to express KRAS ectopically, Libert et al have used 293T cells to study the formation of SGs by KRAS. Grabocka et al show 5-fold difference in SG formation between mut and wt cells, and Libert et al found only a 1.5-fold difference.

We have now performed the experiment in HeLa cells. This result is presented in Fig 3B. We found no effect of mutant KRAS overexpression on stress index in these cells. Our conclusions remain unchanged.

2. The authors did not find 15-d-PGJ2 induced SG as reported previously. Their observations can be interpreted as "Prostaglandin mediated SG formation context-dependent". Hence Libert et al should use multiple cell lines including the one shown by Grabocka and Bar-Sagi, to support their claim that prostaglandin is not involved in SG formation.

Grabocka and Bar-Sagi used the DLD1 KO line (wild-type KRAS line) to test the effects of 15-d-PGJ2 on the stress index. They observed a more than 5-fold increase in the stress index in the presence of 15-d-PGJ2 (compare Figures 1D and 3C in Grabocka and Bar-Sagi).

We tried to carry out the experiment using the DLD1 KO line, but it is not available through cell repositories, and we were unable to identify laboratories that could provide it to us. Consequently, we conducted the experiments on 4 wild-type KRAS lines. This result was shown in Fig 3A of our original manuscript, but this panel included too much information, which made it unclear. For this reason, this result is now shown in a new panel (Fig 3D). It indicates that in the 4 cell lines tested, 15-d-PGJ2 has no effect on the stress index.

3. Non-cytoplasmic stress granules. In the current study, the authors have compared obese and non-obese patients and show that the SGs are present on nuclear and plasma membranes in human samples. The authors also point out that Fonteneau et al show nuclear SGs in their published study. They argue that SGs should be cytoplasmic, not nuclear. However, these observations don't invalidate the presence of SGs in human patient samples. Nuclear stress granules have been reported in several publications. Hence it is not clear if the authors conclude that the SGs shown by them or by Fonteneau et al are artifacts or irrelevant. The authors should explain elaborately.

From the colabeling experiments carried out in Fig 5, we observed 1) that the expression of G3BP1 was mainly cytoplasmic in human PDAC cells, and 2) that in some areas, condensates expressing G3BP1 were present on the plasma and nuclear membranes of PDAC cells. We have never been able to observe the presence of G3BP1 in cytoplasmic SG. From our conclusions, we note that the G3BP1 labeling in Fig 7J of Fonteneau et al also shows the presence of G3BP1-positive condensates 1) on DAPI-positive structures and 2) which are aligned (multiple dots succeeding one another on a line) between two DAPI-positive structures. From there, we indicate that the G3BP1-positive condensates they observe are probably found on cell nuclei and on plasma membranes (which separate two PDAC cells). We conclude that all these observations do not support that cytoplasmic SG are present in PDAC cells. Therefore, we do not believe that the G3BP1 labeling shown by Fonteneau et al is

artificial or irrelevant, but rather that they misinterpreted their results, as they did not colabel G3BP1 with plasma or nuclear membrane markers.

In this paragraph, we talk about “nuclear condensates”, and never use the term “nuclear SG”, to avoid confusion. Nuclear SG have been described in 14 studies (in PubMed); none of these mention the presence of G3BP1 in these nuclear SG, which suggests that they are functionally different from cytoplasmic SG. For this reason, we prefer not to develop an issue related to nuclear SG in our manuscript.

We hope that these explanations will have enabled the referee to better understand the description of our results and the analysis that we make of the results of Fonteneau et al.

4. Antibody issue: the authors claim that the human G3BP1 antibody (611126) doesn't recognize mouse isoform (Fig EV2B). As that antibody was used by Grabocka and Bar-Sagi, Cell, 2016 to prove the presence of SGs in mouse tumors, they question the validity of the data presented by Grabocka et al. However, several points need to be taken into consideration in this context.

a) It seems like the exogenous mouse G3BP1 is expressed at a very low level (verified by other antibodies used in Fig EV2B). It may be possible or G3BP1 (611126) antibody gives a weak signal and is not apparent due to image processing with high contrast (Fig EV2B). Multiple exposures may have helped.

We thank the referee for her/his comment. As suggested by the referee, we had at the time made a longer exposure of the WB presented in Figure EV2B with the G3BP1 antibody (611126). We provide this result below, as material for the referee. The short and long exposures allowed us to conclude that this antibody does not recognize mouse G3BP1.

Lane 1, untransfected HeLa cells; lane 2, HeLa cells transfected with a GFP-expressing vector; lane 3, HeLa cells transfected with a mouse G3BP1-expressing vector. The Western blot membrane was exposed for 5 seconds (left) or 1 minute 40 seconds (right). The G3BP1

antibody (611126) recognized the human form of G3BP1 present in HeLa cells, but not the mouse form expressed by the expression vector, unlike the two other G3BP1 antibodies tested (13057-2-AP and A302-033A. See also Fig EV2B).

b) Some antibodies may not be suitable for immunoblot but can show a signal in IF or IHC. In Fig EV2B, the authors have shown that G3BP1 antibody (A302-033A) recognizes mouse G3BP1. This antibody was also used by Fonteneau et al, to show the existence of mouse PanINs (see methods). Hence the argument is not strong enough to refute the previous findings.

About the G3BP1 antibodies that we used in our study, given the importance of these reagents for a study of the type carried out here, we sought from the start of our work to validate the antibodies in accordance with the highest current standards. These standards involve testing antibodies under conditions of overexpression and inactivation of targeted proteins (see for example Goodman, J Cell Sci 131, 216416, 2018 or <https://www.proteinatlas.org/about/antibody+validation>). Thus, in addition to the Western blot results in Fig EV2B showing overexpression of G3BP1, we also tested the antibodies in a mouse model of conditional inactivation of G3BP1.

To support our claim, we now provide as material for the referee the validation of the A302-033A antibody using the G3BP1 conditional knockout mouse model. We also tested the ability of the A302-033A antibody to detect G3BP1-positive granules, after induction of a heat shock, in a manner identical to what we carried out in Fig 1F for the antibody 13057-2-AP. As a reminder, we were unable with this antibody to detect SG in PanIN of ElaK mice treated with cerulein (in the absence of heat shock).

A

B

A. Immunolabeling carried out on pancreas sections from ElaK (top pictures) and ElaK G3BP1^{f/f} (bottom pictures) mice treated with cerulein, using G3BP1, β -catenin, and amylase antibodies. Cell nuclei were labeled with Hoechst. In ElaK+cerulein mice, all PanIN cells strongly express G3BP1 and β -catenin, while amylase expression is lost. In ElaK G3BP1^{f/f}+cerulein mice, some PanIN lose G3BP1 expression while continuing to express β -catenin (dashed white line), while other PanIN continue to express G3BP1 (dashed yellow line). This is explained by the fact that CreER only recombines part of the lox sites associated with the G3BP1 gene. B. Immunolabeling carried out on pancreas sections from ElaK mice treated with cerulein, shocked at 43°C for 30 minutes. using G3BP1, β -catenin, and amylase antibodies. Cell nuclei were labeled with Hoechst. Among PanIN cells, which show high expression of β -catenin, some cells show the presence of G3BP1-positive granules (inset).

As pointed out by the referee, Fonteneau et al shows the presence of SG in PanIN using this same antibody. We recognize that this labeling resembles SG and added this reference in the corresponding point (page 6). Importantly, it should be noted that Fonteneau et al did not carry out any validation of their immunolabeling conditions, and that in the absence of these validations, artifactual labeling cannot be excluded. An alternative explanation that emerged from a discussion with Grabocka and Bar-Sagi is the presence of an additional unidentified stressor in their mouse, which would not be present in our mice; we had already mentioned this point to some extent in the original version of our manuscript by writing “We concluded that SG are not detectable in mouse PanIN under the usual induction conditions” (page 6). Without a clear explanation of the differences that we observe between our two studies on this particular point, we have taken up the statement of the referee who concluded that the presence of SG in KRAS mutant cancers is not universal. This has now been added in the Abstract (page 2) and on page 10.

Dear Dr. Jacquemin,

Thank you for the submission of your revised manuscript to our editorial offices. I have now received the reports from the referee that I asked to re-evaluate the study, you will find below. As you will see, the referee now fully supports the publication of the study in EMBO reports.

Before we can proceed with formal acceptance, I have these editorial requests I ask you to address in a final revised manuscript:

- Could we have a more active title with not more than 100 characters (including spaces). E.g.:
Stress granules are not present in Kras mutant cancers and do not control tumor growth

- Please provide individual production quality figure files as .eps, .tif, .jpg (one file per figure), of main figures and EV figures. Please upload these as separate, individual files upon re-submission (without their legends).

- Please order the manuscript sections like this, using these names:

Abstract - Keywords - Introduction - Results & Discussion - Methods - Data availability section - Acknowledgements - Disclosure and Competing Interests Statement - References - Figure legends - Expanded View Figure legends

- We now use CRediT to specify the contributions of each author in the journal submission system. CRediT replaces the author contribution section. Please use the free text box to provide more detailed descriptions and do NOT provide your final manuscript text file with an author contributions section. See also our guide to authors:
<https://www.embopress.org/page/journal/14693178/authorguide#authorshipguidelines>

- Please move the tables (cell lines and antibodies) from the methods section to the reagents and tools table (see below).

- All Materials and Methods need to be described in the main text using our 'Structured Methods' format, which is required for all research articles. According to this format, the Materials and Methods section should include a Reagents and Tools Table (listing key reagents, experimental models, software, and relevant equipment and including their sources and relevant identifiers), uploaded as separate file, followed by a Methods and Protocols section in which we encourage the authors to describe their methods using a step-by-step protocol format with bullet points, to facilitate the adoption of the methodologies across labs. More information on how to adhere to this format as well as downloadable templates (.doc) for the Reagents and Tools Table can be found in our author guidelines (section 'Structured Methods'):

- Please make sure that the number "n" for how many independent experiments were performed, their nature (biological versus technical replicates), the bars and error bars (e.g. SEM, SD) and the test used to calculate p-values is indicated in the respective figure legends. Please also check that all the p-values are explained in the legend, and that these fit to those shown in the figure. Please provide statistical testing where applicable. Please avoid the phrase 'independent experiment', but clearly state if these were biological or technical replicates. Please also indicate (e.g. with n.s.) if testing was performed, but the differences are not significant. In case n=2, please show the data as separate datapoints without error bars and statistics. See also:
<http://www.embopress.org/page/journal/14693178/authorguide#statisticalanalysis>

If n<5, please show single datapoints for diagrams. It seems, many diagrams are presently missing statistics and or 'n.s.'. Moreover:

- Please provide the exact p values in the legend of figure 4d.

- Please note that the box plots need to be defined in terms of minima, maxima, centre, bounds of box and whiskers, and percentile in the legend of figure 3e.

- Please note that information related to n is missing in the legend of figure 3d.

- Please note that the error bars are not defined in the legend of figure 3d.

- Please add to each legend (main and EV figures, where applicable) a 'Data Information' section explaining the statistics used or providing information regarding replicates and scales. See:

- Please add scale bars of similar style and thickness to microscopic images, using clearly visible black or white bars (depending on the background). Please place these in the lower right corner of the images themselves. Please do not write on or near the bars in the image but define the size in the respective figure legend. Presently, several scale bars have are too thin. Please check.

- Tables 1 and 2 are datasets. Please upload these as dataset files and add a legend for these on the first TAB. Please name

these Dataset EV1 and Dataset EV2 and change their callouts accordingly.

- Please remove the referee token (and the other login information) from the data availability section and add a direct link (URL) to access the dataset. Please make sure that this dataset is public latest on the day of online publication of the manuscript.
- Please make sure that all the funding information is also entered into the online submission system and that it is complete and similar to the one in the acknowledgement section of the manuscript text file. Presently the FRS-FNRS grant 7.8503.20 seems missing in the submission system. Please check.
- Thank you for providing the requested source data. Please upload this as one folder per figure (with all files for one figure in one folder and ZIPed).

In addition, I would need from you:

Best,

Referee #2:

The authors have adequately addressed this reviewer's concerns.

All editorial and formatting issues were resolved by the authors.

Patrick Jacquemin
Louvain University
Avenue Hippocrate
Belgium

Dear Dr. Jacquemin,

I am very pleased to accept your manuscript for publication in the next available issue of EMBO reports. Thank you for your contribution to our journal.

Yours sincerely,
